# Mechanical Properties of Silicon Carbide Composites Reinforced with Reduced Graphene Oxide

**DOI:** 10.3390/ma17133370

**Published:** 2024-07-08

**Authors:** Kamil Broniszewski, Jarosław Woźniak, Tomasz Cygan, Dorota Moszczyńska, Andrzej Olszyna

**Affiliations:** Faculty of Materials Science and Engineering, Warsaw University of Technology, ul. Wołoska 141, 02-507 Warsaw, Poland; kamil.broniszewski@pw.edu.pl (K.B.); jaroslaw.wozniak@pw.edu.pl (J.W.); tomasz.cygan@pw.edu.pl (T.C.); dorota.moszczynska@pw.edu.pl (D.M.)

**Keywords:** composites, silicon carbide, mechanical properties

## Abstract

This article presents research on the influence of reduced graphene oxide on the mechanical properties of silicon carbide matrix composites sintered with the use of the Spark Plasma Sintering method. The produced sinters were subjected to a three-point bending test. An increase in flexural strength was observed, which reaches a maximum value of 503.8 MPa for SiC–2 wt.% rGO composite in comparison to 323 MPa for the reference SiC sample. The hardness of composites decreases with the increase in rGO content down to 1475 HV10, which is correlated with density results. Measured fracture toughness values are burdened with a high standard deviation due to the presence of rGO agglomerates. The K_IC_ reaches values in the range of 3.22–3.82 MPa*m^1/2^. Three main mechanisms responsible for the increase in the fracture toughness of composites were identified: bridging, deflecting, and branching of cracks. Obtained results show that reduced graphene oxide can be used as a reinforcing phase to the SiC matrix, with an especially visible impact on flexural strength.

## 1. Introduction

Silicon carbide is a very popular material due to its high mechanical, electrical and thermal properties. It is characterized by excellent thermal and chemical stability. It possesses a high hardness value and Young modulus [1]. Its broad spectrum of applications contains supercapacitors [2], high–temperature semiconductors [3], nuclear reactors [4], and the automotive industry [5]. Silicon carbide is very hard to sinter due to its covalent bonds and, in order to enhance the consolidation process, the addition of sintering aids is a necessity. Carbon usage as a sintering activator was first described by Prochazka and was followed by other scientists [6,7]. Its presence reduces the amount of diffusion through the gas phase during sintering, which is a very unfavorable mass transport mechanism. Petrus et al. examined the influence of the morphology of used carbon on the sinterability of silicon carbide [8]. Other than carbon, effective sintering aids come in the form of boron and oxides, like Al_2_O_3_, ZrO_2_, and Y_2_O_3_. Their main role is very different from carbon and is the formation of a liquid phase during the sintering process [9].

Spark Plasma Sintering (SPS) is an advanced technique that utilizes direct current pulses. This method facilitates a more uniform temperature increase throughout the entire volume, compared to conventional sintering methods, by allowing the current to flow through both the die and the powder. During the process, sparks generated between powder particles cleanse the particle surfaces, enhancing sinterability and enabling the use of lower sintering temperatures. SPS also supports rapid heating rates (exceeding 1000 °C/min) and shorter dwell times than traditional sintering methods. This technique is employed to achieve high-density sintered materials, including ultra-high-temperature ceramics, ceramic matrix composites with MAX phases, and laminated materials [10,11,12].

One of the biggest flaws of ceramics, and SiC is not an exception, is low fracture toughness. Strong covalent bonds of silicon carbide contribute to the brittleness of this material. The most effective way to minimize this disadvantageous property is to add a reinforcing phase. In the previous article, the authors examined the effect of the addition of reduced graphene oxide (rGO) to a silicon carbide matrix on thermal and electrical properties [13]. It was observed that rGO has a significant impact on those properties. Since there is a scarcity of scientific publications on the mechanical properties of SiC–rGO composites, the authors decided to contribute to this specific field of materials science. In this study, the influence of rGO content on hardness and fracture toughness was measured. Additionally, the 3-point bending tests for produced composites were performed. As of today, in the Scopus database, there are only a few publications containing measurements of the bending strength of SiC–rGO composites.

Reduced graphene oxide is a very interesting material. It is a product of the thermal or chemical reduction of graphene oxide. Since the reduction is not full, i.e., some residual oxygen and other heteroatoms are still present in the lattice, the product of this reduction is not pure graphene but rGO. Reduced graphene oxide possesses properties similar to graphene but is more hydrophilic, which is a big advantage when it comes to sintering silicon carbide-based composites. Higher wettability enhances the sintering and additionally contributes to creating a stronger SiC–rGO interface. Modern industry demand for pure graphene is still growing, but the production methods are still complicated, expensive, and not satisfactory in the output amount. This alone is a good reason to start a search for alternatives, such as reduced graphene oxide, which is cheaper and available in greater amounts. Reduced graphene oxide possesses, like graphene, a very interesting property of aligning itself in a parallel direction to the direction of a force during the sintering process. Because the SPS technique connects the heating of a powder with simultaneous pressing, the rGO particles align in one direction. This phenomenon allows the composites with rGO to exhibit different mechanical, electrical, and thermal properties depending on the direction in which they were examined.

The use of rGO as a reinforcing phase in the silicon carbide matrix composites has many advantages. Huang et al. observed that the addition of rGO to the SiC matrix can improve the fracture toughness of composites [14]. The main mechanisms responsible for the enhancement of the K_IC_ value were crack bridging, crack deflection, and pullout of rGO particles. There were no visible maximums and the increase in rGO content increases the fracture toughness of sinters. This was also observed for other types of ceramic matrixes, such as AlN [15]. A different impact on fracture toughness was observed by Chen et al. [16] when the pure graphene was introduced into a silicon carbide matrix. A visible peak was present for a SiC–1 wt.% GNP composite after which a further increase of graphene content had a negative impact on the K_IC_ values. They have also noted a constant decrease in hardness with the increase in GNP content. The same trend in hardness values was observed by Huang et al. [17] but with the addition of reduced graphene oxide. It was reported that an addition of rGO to the SiC matrix may be even more effective in terms of increasing fracture toughness than the addition of GNPs and the differences in flexural strength values are very similar [18,19].

The reduced graphene oxide belongs to the family of graphene-like materials and its properties stand in between the pure graphene and graphene oxide. It possesses high mechanical properties and exhibits a hydrophilic characteristic, which makes it a promising reinforcing phase to the silicon carbide matrix. Despite having potential, there are still some fields of research that need attention. There is a shortage of scientific articles covering the mechanical properties of SiC–rGO composites, and in particular, flexural strength, which was the reason for choosing this field of materials science as the main focus of this study.

## 2. Materials and Methods

The SiC–rGO composites were created with the use of powder metallurgy methods [20]. The powder substrates were all commercially available powders presented in Table 1.

The preparation of the powder mixtures was carried out for 24 h in propan-2-ol using a horizontal-type ball mill. Powders were then dried and sieved (# 250 μm). The Spark Plasma Sintering method was used to consolidate all of the composites (HP-D10, FCT Systeme GmbH, Effelder-Rauenstein, Germany). The parameters were as follows: T = 2000 °C, t = 1 h, *p* = 50 MPa, vacuum, and heating and cooling rate = 250 °C/min. The SPS pulse on/off ratio was set to 15:1 ms. The series of eight composites with various reduced graphene oxide content was produced: 0.25/0.5/0.75/1/1.5/2/2.5/3. In addition, a reference SiC sample was sintered. A constant content of 0.3 wt.% boron was used in all samples as a sintering aid. The reference sample was prepared using 0.3 wt.% boron and 0.3 wt.% carbon black as sintering aids. To achieve a high-density SiC sample through the SPS process under the same sintering parameters as the composites, the use of sintering aids was essential. This approach allowed us to prepare the reference sample without the need to increase the temperature or extend the dwell time, thereby maintaining consistency in phase composition and microstructure. The Archimedes method was used to measure the relative density of produced composites based on the theoretical density calculated using the rule of mixtures. The powder morphology, fracture surface, and crack propagation of sinters were observed using scanning electron microscopy (SEM, HITACHI S5500, Tokyo, Japan). The fracture surface after the 3-point bending test was examined using SEM (Axia ChemiSEM, Thermo Fisher Scietific, Waltham, MA, USA). Phase analysis was carried out on an X-ray diffractometer (Bruker D8 Advance, Billerica, MA, USA). A dispersive Raman spectrometer with a 532 nm laser length was used to analyze reduced graphene oxide powder and rGO in sintered composites (Nicolet Almega, Thermo Fisher Scietific, Waltham, MA, USA). The Vickers indentation technique was used to the measure hardness (HV10) and fracture toughness (K_IC_) of composites (FV–700e). The fracture toughness was calculated based on the Niihara-Morena-Hasselman [21] Equation (1).
K_c_ = 0.0711(H_v_ × a^1/2^)(E/H_v_)^2/5^(c/a)^−3/2^(1)

H_v_—Hardness

E—Young’s modulus

c—half of the diagonal of Vickers indent plus crack length

a—half of the diagonal of Vickers indent

The Young modulus values were measured with the use of an ultrasonic method (Optel refractometer). Flexural strength was determined in a three-point bending test (MTS Exceed E43). The samples were cut into rods with dimensions of 3 × 4 × 45 mm according to the standard PN–EN 841–1 [22]. The traverse moving speed was set to 0.5 mm/s to ensure that cracking occurs in a 5–15 s window. The rods were cut in a way that ensures that the direction of a load in the bending test is perpendicular to the alignment direction of the reduced graphene oxide particles. This allows us to perform the test in conditions where the produced composites exhibit their optimal performance and the effect of rGO particles on flexural strength is the highest.

## 3. Results

The produced silicon carbide matrix composites with the reduced graphene oxide addition are characterized by their high relative densities, with the highest value of 99.5% for SiC–0.75 wt.% rGO sinters (Figure 1). It was observed that the presence of a reinforcing phase below 0.75% caused an increase in the degree of sinter consolidation. The addition of rGO above 1.5% caused a slight decrement in values of density down to 97.6% for the SiC–3 wt.% rGO. The reference SiC sample achieved a densification of 98.5%. The sintering curves for the representative SiC–2 wt.% rGO composite are presented in Figure 2. The entire process was divided into five segments. The second segment begins at 250 °C, the third at approximately 1180 °C, the fourth at the dwell temperature of 2000 °C, and the final segment encompasses the cooling of the sinters. The microstructure observations revealed the presence of elongated grains of silicon carbide (Figure 3a). A strong interface of SiC–rGO in the produced composites can be observed. The agglomerates of reduced graphene oxide can also be spotted in the composite microstructure (Figure 3b).

Figure 4 shows an X-ray diffraction pattern for the exemplary composite containing 2 wt.% rGO. The presence of three phases was detected in consolidated samples: SiC (6H polytype), rGO, and SiO_2_. The Raman spectroscopy results are presented in Figure 5. Reduced graphene oxide powder shows a characteristic series of bands with D and G peaks positioned at 1334 cm^−1^ and 1576 cm^−1^, respectively. Additionally, the 2D and D+D’ bands were observed at 2672 cm^−1^ and 2916 cm^−1^, respectively. The I_D_/I_G_ calculated ratio for rGO powder equals 1.15. The produced composites are characterized by bands indicating the presence of carbon structures. The bands are shifted towards higher values: G—1355 cm^−1^, D–1590 cm^−1^, 2D–2700 cm^−1^, D+D’–2956 cm^−1^. In the examined sinters, the D’ band at a position of 1619 cm^−1^ may also be observed. The I_D_/I_G_ ratio for the exemplary composite equals 0.65 and is much lower than the one calculated for the reduced graphene powder substrate.

The measured Young’s modulus for the produced composites is presented in Figure 6. The highest value, 397 GPa, was observed for the SiC–0.25 wt.% rGO composite, which is slightly higher than the 388 GPa measured for the reference sample. A noticeable trend of decreasing Young’s modulus with increasing rGO content is observed. The lowest value, 355 GPa, was measured for the SiC–3 wt.% rGO composite.

Figure 7 shows the changes in hardness for obtained sinters. It can be noted that the increase in the content of reduced graphene oxide in the silicon carbide matrix leads to a decrease in measured hardness values. The addition of low content of rGO, below 1%, only slightly impacts the hardness, which diminishes by 7.5% in comparison to the pure SiC sinter. The lowest value is acquired for the SiC–3 wt.% rGO sample and equals 1475 HV10. The measured fracture toughness presented in Figure 8 is marked with a high standard deviation and reaches values in the range of 3.22–3.82 MPa*m^1/2^. The reference sample has a K_IC_ value of 3.26 MPa*m^1/2^. Figure 9 presents the characteristics of crack propagations in produced composites. The main observed mechanisms for enhancing the fracture toughness in produced composites were crack deflection (Figure 9a), various cracks bridging (Figure 9c–e), and crack branching (Figure 9e). The occurrences of cracks ending inside the rGO agglomerates were also observed (Figure 9b).

Figure 10 shows flexural strength results for produced sinters. The addition of up to 0.5 wt.% rGO caused almost no changes in the flexural strength of the composites compared to the pure SiC sample value of 332 MPa. It can be observed that further increasing the reduced graphene oxide content enhances the flexural strength. The highest value was obtained for the 2 wt.% rGO sample and equals 503.8 MPa. The fracture images after the flexural test are shown in Figure 11a–d. The fractures’ origins were located on the convex side of samples where the tensile strains are located. Compression curls were observed in the concave side of beams where the compression strains reside. Some samples were fractured into more than two parts.

## 4. Discussion

The sintering process parameters led to obtaining the high-density composites. It concluded that the sintering process was carried out with the proper parameters. Silicon carbide is a material that is hard to sinter because of the nature of its chemical bonds. The addition of phases that create a liquid phase is almost a necessity. In this work, boron was chosen due to its effectiveness and the authors’ previous experiences with this phase [23,24]. The second type of sintering aid used in the silicon carbide sintering process is carbon. This work treats the rGO not only as a reinforcing phase but also as a source of carbon to enhance the sintering. This is visible in the increase in density for composites in comparison to the pure SiC sample. The effect of the addition of carbon in the form of reduced graphene oxide increases the relative density from 98.6% to 99.3%. The slight decrease in density for composites above 1.5 wt.% rGO content is due to the higher amount of rGO agglomerates. It was observed that the agglomerates of the reduced graphene oxide contain pores between the platelets and thus the higher their content is, the higher the porosity of the composite is.

The first segment of the sintering process involves waiting for the temperature to reach 250 °C, the lowest temperature detectable by the installed pyrometer, which explains the flat temperature line. In the second segment, a slight negative piston movement occurs due to the thermal expansion of the graphite die and punches, while the temperature remains too low to induce densification. The third segment marks the beginning of powder densification through mass transfer and neck formation, evident by a substantial relative displacement of the piston and an increase in punch speed. The fourth segment coincides with the sintering dwell time, where the sintering curve flattens, indicating that the dwell time is sufficient and a high rate of densification has been achieved. The final segment involves the cooling of the samples, set at a speed of 250 °C/min to prevent cracking due to thermal stresses.

Microstructure observations showed the presence of elongated SiC grains, characteristic of a hexagonal phase. This implies that a phase transformation occurred and the regular SiC phase changed to the hexagonal one. The polytypism of silicon carbide is a very complex topic since there are over 250 known polytypes [25]. Fortunately, there are only a few common ones. Moreover, at the sintering temperature of 2000 °C, there can exist only four types of SiC polytypes. The only regular polytype is 3C and there are three hexagonal polytypes named in order of the chance of occurrence: 6H, 4H, and 8H [26]. The phase transformation of a regular β–SiC at high temperatures is common as it is an unstable phase and, as a result, the stable hexagonal α–SiC phase is created [27]. The observations of a composite microstructure also revealed the strong interface between the SiC matrix and rGO particles. A trace number of pores was present on the SiC–rGO interface and the majority of them were located inside the rGO agglomerates.

The produced composites consist of two main phases, SiC and rGO, and a trace amount of SiO_2_. The presence of silicon oxide is related to the thin layer it creates on silicon carbide particles. The XRD analysis results are similar to what other researchers reported [28,29]. The polytype of silicon carbide was defined as 6H, which confirms the phase transformation. It is known that some elements may influence the stabilization of certain SiC polytypes. It was reported that the addition of boron tends to stabilize the 6H polytype and since all samples contain boron, the formation of this particular polytype could be expected [30].

Raman spectroscopy was used to determine possible changes to the reduced graphene oxide during the sintering process. It was observed that peaks from rGO in composites are slightly shifted towards higher values, which is correlated to the stacking of rGO. The peak at 1619 cm^−1^, in the produced composite, was identified as the D’ band, which is correlated with the level of disorder. The D’ band is also present in the Raman spectra of the rGO powder, but it overlaps with the G band. The shift to higher frequencies for the rGO in composite implies a decrease in the number of defects. The calculated values of the I_D_/I_G_ ratio are lower for reduced graphene oxide peaks in composites, which means a decrease in disorder, which correlates well with the D’ band shift. This would suggest that during the sintering process, a further reduction of rGO occurred, which is connected to the reduction of the residual oxygen and thus a decrease in the rGO level of defects. Reduction of rGO was reported by Liu during sintering with the use of the High-Pressure High-Temperature technique [31].

An increase in rGO content corresponds to a decrease in density, which results in lower measured values of the Young’s modulus. These measured values correlate well with the relative density of the composites. Additionally, the decrease in the Young’s modulus can be attributed to the lower Young’s modulus of rGO compared to the SiC matrix. While the Young’s modulus of graphene can reach approximately 1 TPa, the reduced graphene oxide (rGO) is only about 250 GPa [32,33]. The lower Young’s modulus observed in the reference sample, compared to the SiC–0.25 wt.% rGO composite, is influenced by its lower density.

The addition of reduced graphene oxide to a silicon carbide matrix leads to a decrease in measured hardness. The results correlate well with the other researchers’ observations regarding the impact of graphene family fillers on composite hardness [15,16,17]. The enhancement of hardness may be obtained through the microstructure modification provided by the introduction of graphene-like phase but this often leads to the increased porosity, which can weaken this enhancing effect. The microstructure refinement effect on the hardness of the composites can also be countered by the presence of agglomerates. In the produced composites, it was observed that the higher the rGO content is, the lower the hardness becomes.

Fracture toughness results are characterized by high standard deviation and the overall trend of the measured values is not easy to define. Other researchers reported an increase in fracture toughness caused by the introduction of the rGO into the SiC matrix [16,34]. The reported increase in K_IC_ values has a unimodal or close-to-linear character for composites with low rGO content. Measured in this publication, values differ only slightly for composites with rGO content below 2%. The composite with 2 wt.% rGO has the highest fracture toughness among all examined samples and above that rGO content, the measured values decrease. Figure 9 shows the main reason for the high standard deviation values for the fracture toughness results. The randomness in shape and size of agglomerates may result in different values of the fracture toughness measurements. Scanning electron microscopy allowed us to observe the mechanisms responsible for the increasing K_IC_ value. Figure 9a shows an example of crack deflection. Very rarely an ending of crack inside the rGO agglomerate was spotted, as seen in Figure 9b. Many different variations in crack bridging are presented in Figure 9c–e. The crack branching could also be observed in produced composites (Figure 9f). Although three main enhancing fracture toughness mechanisms were identified (deflection, bridging, and branching), they are referring not to the single rGO particles but rather to the agglomerates. The particles inside agglomerates are not strongly connected and the cracks, due to the platelet shape of rGO, tend to propagate along the planes. The effectiveness of the above-mentioned mechanisms is thus reduced. This would explain the fracture toughness values for the five composites with rGO below 2 wt.%. The fracture toughness-increasing mechanisms are present but have lower effectiveness due to the agglomeration of rGO particles, thus resulting in almost no impact on K_IC_ values. Above 2 wt.% rGO, the decrease in K_IC_ values is also related to the decrease in density of produced composites.

An increase in rGO content in the SiC matrix leads to an increase in the flexural strength of the produced composites. The flexural strength values for up to 0.5 wt.% rGO content are very similar to each other. The rise can be observed afterward and shows an influence of the addition of the reinforcing phase. The beams were cut in a way that rGO particles are aligned perpendicular to the direction of the load in the bending test. This allows the rGO flakes to have the optimal orientation and, in consequence, the highest impact on the flexural strength of the composites. A slight drop in measured values can be observed for the SiC–2.5% wt. rGO and SiC–3% wt. rGO, which is correlated with the lower density for these two sinters.

The comparison of mechanical properties of SiC matrix composites reinforced with various materials from the graphene family is presented in Table 2. Despite the presence of the rGO agglomerates in the microstructure of the produced composites, the mechanical properties demonstrate significant potential. This underscores the promise of further research and optimization of these composites.

## 5. Conclusions

The Spark Plasma Sintering (SPS) method has been successfully used to manufacture high-density SiC–rGO composites with a relative density exceeding 97.5%. The key findings of this study are as follows:Sintering Challenges and Additives:Silicon carbide (SiC) is inherently difficult to sinter, necessitating the use of sintering aids.Boron was chosen as a sintering activator, facilitating the liquid sintering process.Reduced graphene oxide (rGO) served dual roles as a reinforcing phase and as a carbon source.Phase Transformation and Microstructure:Due to the instability of the β–SiC phase, a phase transformation occurs during the sintering process at 2000 °C.XRD analysis confirmed the formation of hexagonal α–SiC and the presence of elongated grains in the microstructure.The produced composites comprise three phases: α–SiC (6H), rGO, and SiO_2_.Raman Spectroscopy Analysis:Raman spectroscopy revealed a shift in the characteristic peaks for rGO towards higher values in the composites compared to the rGO powder.This shift indicates a lower number of defects in the rGO within the sintered samples.The reduced ID/IG ratio and the appearance of a D’ band in the composites further confirm the decrease in rGO defects, attributable to further reduction during the sintering process.Mechanical Properties:The addition of GO to the SiC matrix resulted in a decrease in hardness, consistent with findings reported by other researchers [15,16,17].Fracture toughness measurements showed a high standard deviation due to the presence of rGO agglomerates in the microstructure.Identified toughening mechanisms, such as crack deflection, bridging, and branching, are less effective due to the loosely connected rGO particles within the agglomerates.The addition of rGO to the SiC matrix resulted in a decrease in the Young’s modulus values, which correlates well with density results.Despite SEM observations indicating a strong SiC–rGO interface, the presence of agglomerates significantly impacts the physical properties.Flexural Strength and rGO Alignment:The rGO tends to align perpendicularly to the direction of the force applied during the sintering process [36,37].Composite beams were cut such that the rGO particles aligned perpendicular to the load direction applied in the three-point bending test.An increase in the rGO content in the matrix led to an increase in flexural strength.Fracture origins were located on the convex side of the beams, where tensile strains are present.

## Figures and Tables

**Figure 1 materials-17-03370-f001:**
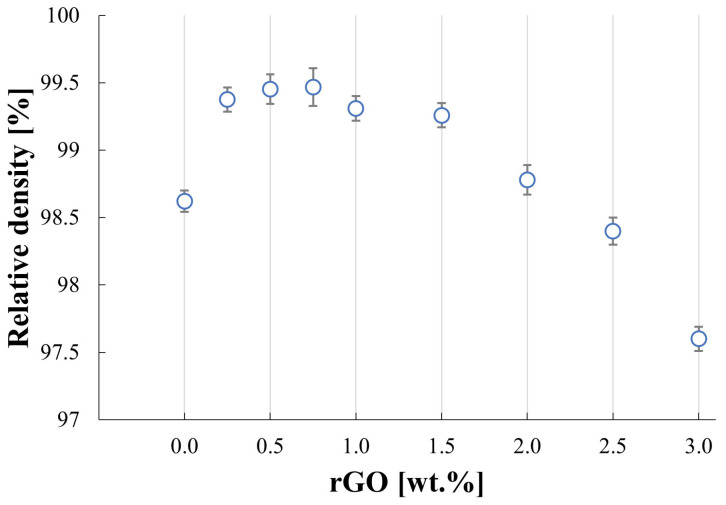
Relative density results for produced SiC–rGO composites.

**Figure 2 materials-17-03370-f002:**
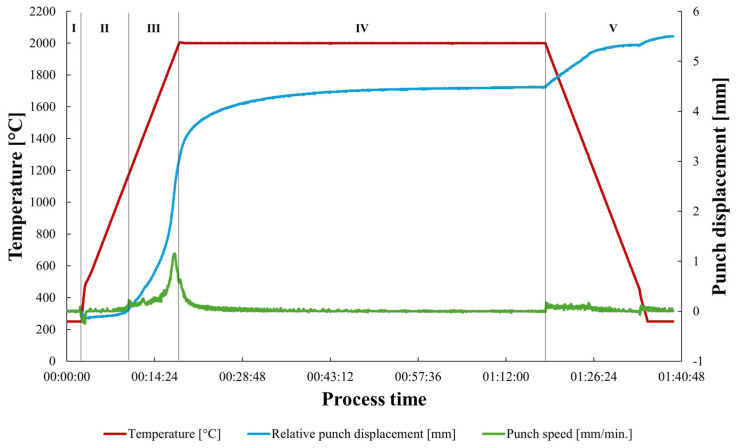
SPS sintering curves for SiC−2 wt.% rGO composite.

**Figure 3 materials-17-03370-f003:**
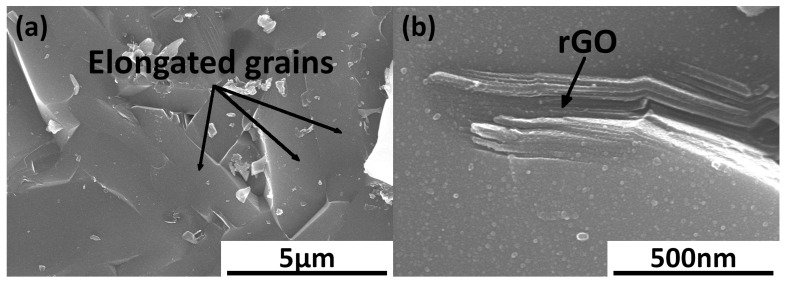
SEM image of surface fracture of SiC–2 wt.% rGO: (**a**) Visible elongated grains of silicon carbide; (**b**) Reduced graphene oxide agglomerate and the interface between SiC matrix and rGO particles.

**Figure 4 materials-17-03370-f004:**
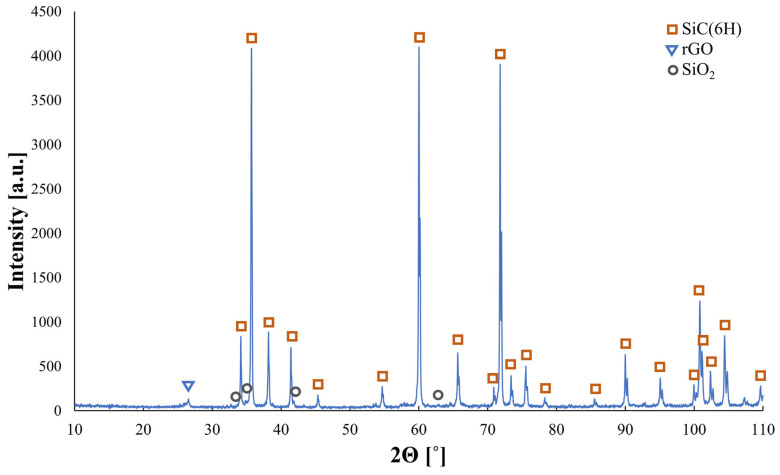
Exemplary phase analysis of composite containing 2 wt.% rGO. Three phases were detected: the matrix phase–SiC (6H polytype), the reinforcement phase–rGO, and a trace amount of SiO_2_.

**Figure 5 materials-17-03370-f005:**
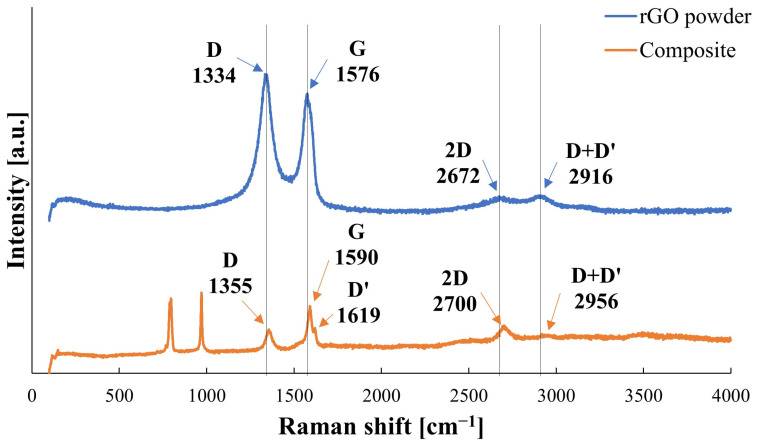
The Raman spectra of pure rGO powder and the representative SiC−2 wt.% rGO composite. Vertical lines were placed in the peaks for the rGO powder substrate to help the observation of a slight shift towards higher values for the sinter.

**Figure 6 materials-17-03370-f006:**
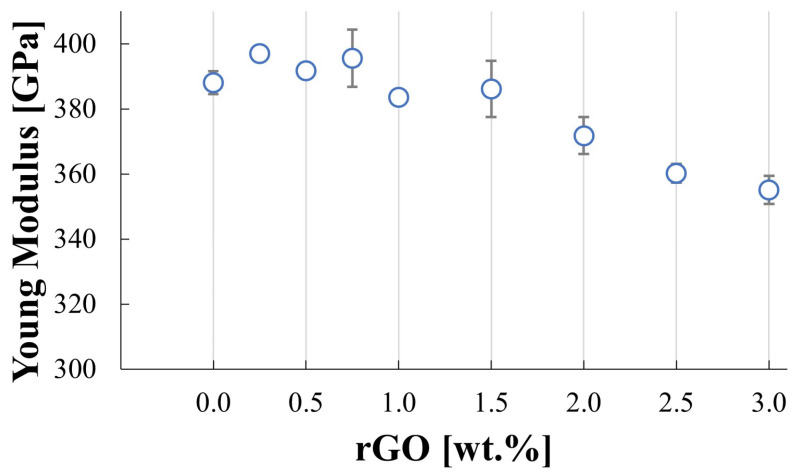
Measured values of Young modulus for sintered composites.

**Figure 7 materials-17-03370-f007:**
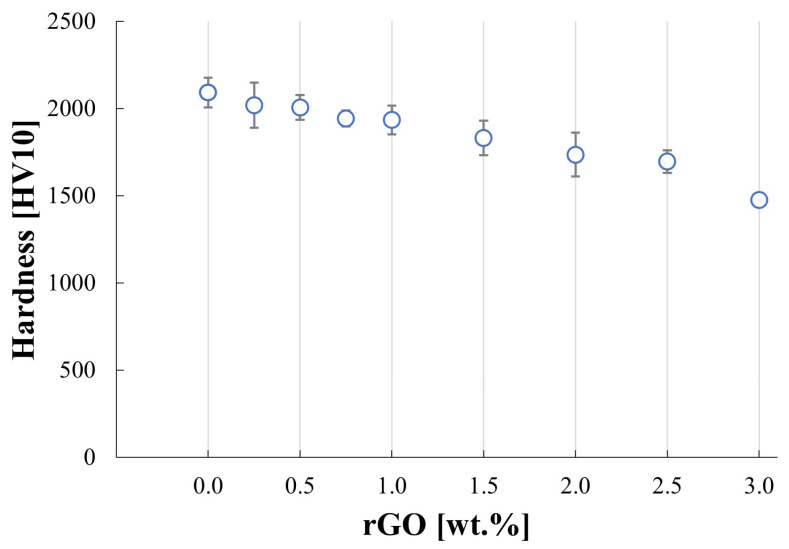
Vickers hardness test results for SiC matrix composites reinforced with rGO.

**Figure 8 materials-17-03370-f008:**
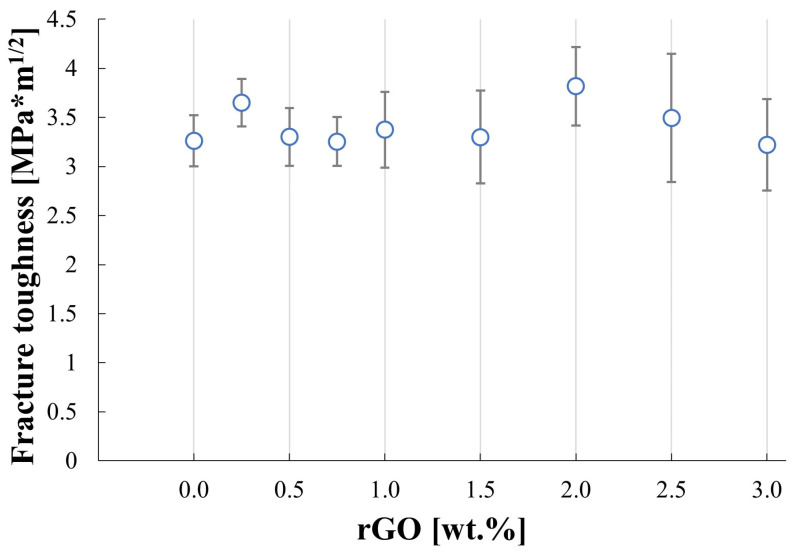
Results of fracture toughness of silicon carbide matrix composites reinforced with reduced graphene oxide.

**Figure 9 materials-17-03370-f009:**
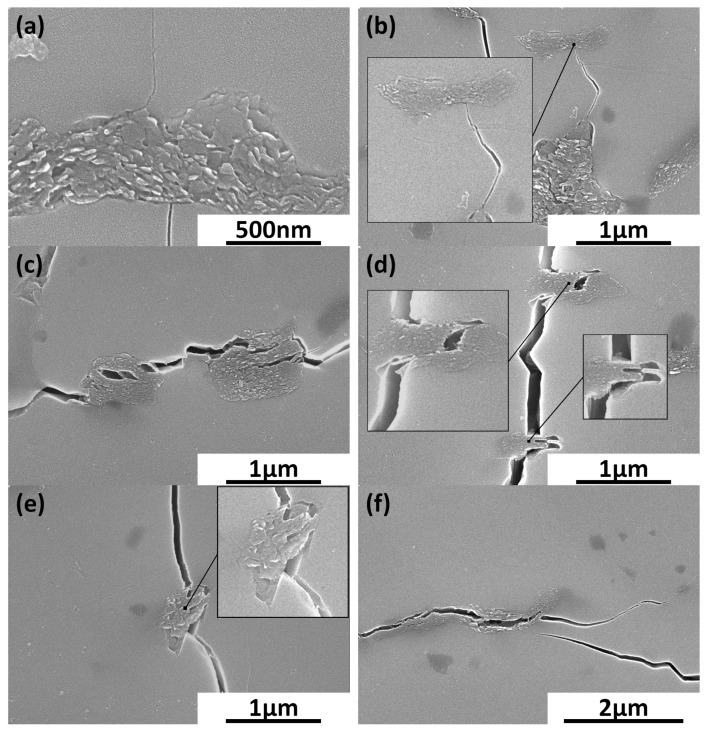
Main mechanisms of increasing the fracture toughness identified in produced SiC–rGO composites. (**a**) Crack deflecting; (**b**) Crack ending; (**c**–**e**) Crack bridging (**f**) Crack branching.

**Figure 10 materials-17-03370-f010:**
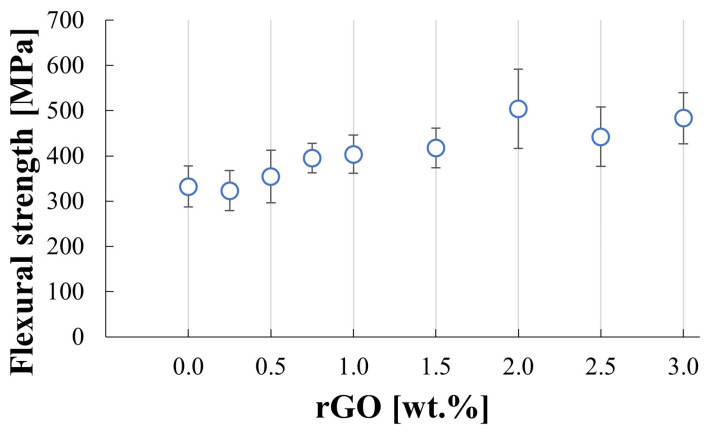
Flexural strength results for produced composites.

**Figure 11 materials-17-03370-f011:**
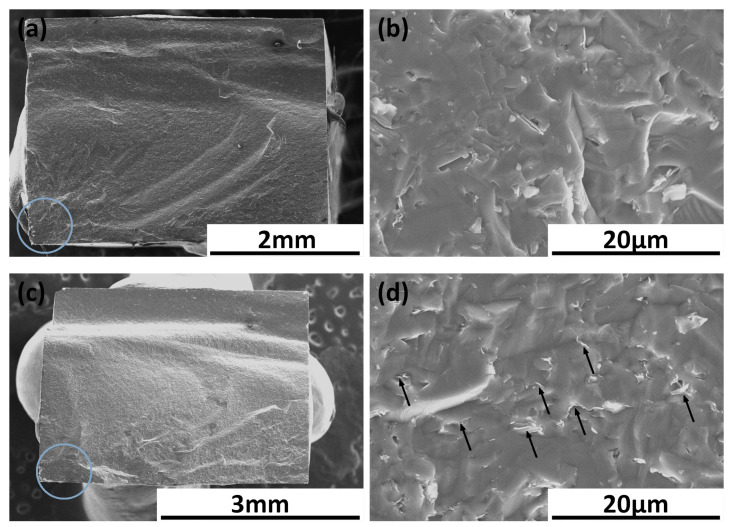
SEM images of fractures of beams after 3–point bending test. The compression curl area is located in the upper part of the samples. The blue circle indicates a possible fracture origin. (**a**,**b**) SiC–0.75 wt.% rGO composite; (**c**,**d**) SiC–2 wt.% rGO composite with marked rGO particles.

**Table 1 materials-17-03370-t001:** Commercial powder substrates used in the powder metallurgy process, leading to the creation of high-density SiC–rGO composites.

Powder	APS ^1^	Manufacturer
Silicon carbide	0.42 µm	Alfa Aesar (Ward Hill, MA, USA)
rGO	<40 µm	Łukasiewicz Research Network (Warsaw, Poland)
Boron	0.39 µm	International Enzymes Limited (Fareham, UK)
Carbon	<100 nm	Sigma-Aldrich (Burlington, VT, USA)

^1^ Average Particles Size.

**Table 2 materials-17-03370-t002:** Comparison of the graphene family phases and their influence on chosen mechanical properties of silicon carbide matrix composites. GNP—Graphene Nano Platelets, GPLs—Graphene Platelets, MLG—Multi–layer graphene.

Composite	Hardness	Fracture Toughness [MPa*m^1/2^]	Flexural Strength [MPa]	Ref
SiC–0.5% wt. rGO	2006 HV10	3.30	323	-
SiC–2% wt. rGO	1735 HV10	3.82	503	-
SiC–1% wt. GNP	26.3 GPa	6.50	560	[16]
SiC–2% wt. GPLs	20.82 GPa	3.70	290	[35]
SiC–1% wt. MLG	26.64 GPa	4.1	-	[23]
SiC–1% wt. GO	-	3.4	239	[34]

## Data Availability

All data included in this study are available upon request by contact with the corresponding author.

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
