# Peer review of "Mechanical Properties of Silicon Carbide Composites Reinforced with Reduced Graphene Oxide"

_materials, 2024, doi:10.3390/ma17133370_

Round 1

Reviewer 1 Report

Comments and Suggestions for Authors

The paper by Broniszewski et al. is devoted to the problem of strengthening SiC-based ceramics by adding graphene oxide. Ceramic composites with varying graphene oxide content were consolidated using the promising spark plasma sintering (SPS) method, and their functional and structural properties were thoroughly studied. Given the relevance of the problem being addressed, the high demand for SiC-based hard materials, and the objective scientific novelty, I recommend this paper for publication in the journal Materials. However, to ensure it meets the highest international standards in this field, a major revision is required. The recommended revisions include:

1. The variation in structural parameters of the consolidated ceramics as a function of relative graphene oxide content (such as: weight fractions of target phases, unit cell parameter values, crystallite sizes and microstrains) should be studied. These characteristics should be calculated using whole-powder pattern fitting methods.

2. Figures 3 and 4 should be modified to show XRD patterns and Raman spectroscopy data as a function of relative graphene oxide content on these plots.

3. For the XRD data, the reference card numbers from the powder diffraction database should be provided, along with a comparative description of the unit cell parameters for the synthesized materials versus the reference standards.

4. To confirm the homogeneity of the composite phase composition, elemental distribution maps obtained by EDS should be included.

5. To describe the processes occurring during SPS, plots of absolute shrinkage and shrinkage rate versus sintering temperature should be provided (e.g. 10.1016/j.jwpe.2024.105042).

6. For additional characterization of processes during SPS, thermal analysis methods such as DSC and TGA may need to be employed.

7. The introduction should be supplemented with a description of the relevance and prospects of using the SPS method to produce functional ceramic materials, with references to recent scientific literature.

8. The conclusion contains an incorrect statement. Line 295 states "This correlates well with what was reported by other researchers". Making this statement without citing the relevant scientific works is incorrect.

9. The conclusion should be revised. All conclusions should be presented in bullet points and in a more structured way. Any facts not directly arising from the investigations carried out in this work should be supported by references from the scientific literature. For example, line 303 states without a proper reference: "the reduced graphene oxide has a tendency to align in the direction perpendicular to the direction of the force applied in the sintering process".

10. The text should include a comparative table showing the functional characteristics of the synthesized ceramics (strength, hardness, etc.) compared to SiC-based ceramics strengthened with additives other than graphene oxide from other scientific works. This will practically demonstrate the prospects of using the graphene oxide strengthening method for SiC.

Reviewer 2 Report

Comments and Suggestions for Authors

1. Four powders were used as the starting materials in Table 1. Boron was used as a sintering aid. What was the role of carbon?

2. The equation adopted to calculate the fracture toughness is needed to be given in the paper. Also, all parameters and measured data used to calculate the fracture toughness in the equation have to be reported in the paper. 

3. It is easy to find in the literature that SiC possesses the fracture toughness of 4.6 MPa m1/2 and flexural strength of 550 MPa. These values are higher than those reported for the SiC-rGO composite of this study. Authors are suggested to discuss or comment on this issue, or to provide explanations for the lower values of fracture toughness and flexural strength.

4. This study aimed to provide the mechanical properties of SiC–rGO composites. Only two properties were measured in this study. It is suggested that more mechanical properties should be required, such as elastic modulus, compressive strength, and Poisson's ratio. 

5. Does the addition of rGO affect the thermal and electric properties of SiC?

Round 2

Reviewer 1 Report

Comments and Suggestions for Authors

The authors have worked through all the remarks and responded to all the comments. The paper can be published in its current form. I wish my esteemed colleagues success in their future studies!

Reviewer 2 Report

Comments and Suggestions for Authors

The revised paper can be accepted for publication.